# Neutral Point Clamped Transformer-Less Multilevel Converter for Grid-Connected Photovoltaic System

P. Madasamy [1,*], Rajesh Verma [2], A. Rameshbabu [3], A. Murugesan [4], R. Umamageswari [5], Josiah Lange Munda [6], C. Bharatiraja [7] and Lucian Mihet-Popa [8,*]

1  Department of Electrical and Electronics Engineering, Alagappa Chettiar Government College of Engineering and Technology, Karaikudi 630003, Tamil Nadu, India
2  Electrical Engineering Department, King Khalid University, Abha 62529, Saudi Arabia; rkishor@kku.edu.sa
3  Department of Electrical and Electronics Engineering, School of EEE, Sathyabama Institute of Science and Technology, Chennai 600119, Tamil Nadu, India; rameshbabu.eee@sathyabama.ac.in
4  Department of Electrical and Electronics Engineering, K S R Institute for Engineering and Technology, Namakkal 637215, Tamil Nadu, India; murugesan.a@gmail.com
5  Department of Electrical and Electronics Engineering, Adhiparasakthi Engineering College, Kalavai 632506, Vellore, India; umaravi2527@gmail.com
6  Department of Electrical Engineering, Tshwane University of Technology, Pretoria 0001, South Africa; MundaJL@tut.ac.za
7  Department of Electrical and Electronics Engineering, SRM Institute of Science and Technology, Chennai 603203, Tamil Nadu, India; bharatiraja@gmail.com
8  Faculty of Engineering, Østfold University College, Kobberslagerstredet 5, 1671 Kråkeroy-Fredrikstad, Norway
*  Correspondence: mjasmitha0612@gmail.com (P.M.); lucian.mihet@hiof.no (L.M.-P.)

**Abstract:** Transformer-less (TL) inverter topologies have elicited further special treatment in photovoltaic (PV) power system as they provide high efficiency and low cost. Neutral point clamped (NPC) multilevel-inverter (MLI) topologies-based transformer-less are being immensely used in grid-connected medium-voltage high-power claims. Unfortunately, these topologies such as NPC-MLI, full-bridge inverter with DC bypass (FB-DCBP) suffer from the shoot-through problem on the bridge legs, which affect the reliability of the implementation. Based on the previous above credits, a T type neutral point clamped (TNP)—MLI (TNP-MLI) with transformer-less topology called TL-TNP-MLI is presented to be an alternate which can be suitable in the grid-connected PV power generation systems. The suggested TL-TNP-MLI topologies free from inverter bridge legs shoot-through burden, switching frequency common-mode current (CMC), and leakage current. The control system of the grid interface with hysteresis current control (HCC) strategy is proposed. The effectiveness of the proposed PV connected transformer-less TNP-MLI topology with different grid and PV scenario has been verified through the MATLAB/Simulink simulation model and field-programmable gate area (FPGA)-based experimental results for a 1.5 kW system.

**Keywords:** photovoltaic system; transformer-less inverter; neutral point clamped multilevel inverter; hysteresis current control; PV tied grid-connected system

## 1. Introduction

Renewable energy sources (RES) are a current asset for the electric power generation. Photovoltaic (PV), fuel cells, and wind turbines are the most popular among the renewable sources. After thumping about 260 GW of PV power installed capacity by the end of 2014, global PV generation power is expected to hit 400 GW soon [1,2]. All PV modules generate parasitic earth capacitance (CPV) approximately 1–2 µF/kW in the middle of the PV modules and their ground. As a result, any PV connected high-frequency inverter/converter suffers from a high leakage current which emanates from the PV system [3,4]. To avoid this problem, a low-frequency isolated transformer is directly associated in the middle of the

PV-tied inverter and grid. Hence PV modules are grounded directly, which guarantees the grid and PV inverter own safety aspects, and output voltage requirements and conquers the photovoltaic dc-component in the grid. By overcoming the transformer usage issue, researchers have proposed different PV-powered transformer-less (TL) inverters for PV-tied grid-connected power system for both single and three phases [5–7].

In this connection, neutral-point-clamped multilevel inverter (NPC-MLI) is broadly espoused in TL PV-tied systems [8–12]. The NPC-MLI topology can defeat the inconvenience of common-mode leakage current which limits the DC component injection in the grid [13]. Three-level NPC-MLIs are vastly used in motor control and PV-grid-connected applications compared to higher level MLIs. The NPC-MLI uses the number of switches and clamping diodes and provides the multiple voltage levels through the different phases. The DC-link capacitor is connected in series and split the DC-link voltage equally to make the voltage levels in the output. The level increases in the NPC-MLI are constructed by increasing the DC-link capacitors, clamping diode, and increasing the switches. The main drawback of the NPC-MLI is maintaining the DC-link capacitor balancing. Hence the NPC-MLI topologies are limited to the three levels [8].

The paper [14] Xiao et al. investigated the detailed analytical model of leakage current for PV-powered grid-connected single-phase transformer-less inverter. With this idea, Gonzalez et al. has deliberated the ideology and characterized elimination of leakage current for PV-grid-connected TL single-phase NPC-MLI system [15]. The prevention of the shoot-through problem on the inverter bridge legs is worth in view of the reliability enhancements to NPC-MLI. The dual-buck half-bridge inverter (DB-HBI) can evade the shoot-through issue, which was proposed in [16]. However, DB-HBI is able to work with the bipolar modulation strategy only, which results in enormous voltage stress upon power devices. In order to resolve these problems, MLI configuration was established [17,18], which is capable of working at higher efficiencies. Based on half-bridge-type TL inverter, Huafeng et al. have proposed PV-grid-associated single-phase split-inductor-based transformer-less NPC-MLI using a variable and fixed hysteresis band with no leakage current and shoot through problems with reasonable transient and steady-state performance [19].

Nevertheless, in this topology operation, the zero vector (0110) and non-zero vector (1100, 0011), the current paths contain two power devices. More importantly, the reported inverter current commutation involves all four IGBTs, which leads to a high power conversion loss and the higher converter switches conduction loss. Though NPC-MLI is much stable for transformer-less PV-based RES, it habitually operates in the range of lower power instead of rated power low/medium power. NPC-MLI looks prominent since their IGBT and diode have comparatively higher state voltage/current ratio in the range of low and medium powers [20–22]. Alternatively, another inverter in NP-MLI family called T shape neutral point MLI called T type NPC (TNP-MLI) is being studied more and more to increase the efficiency of the system [23,24]. Efficiency of the TNP-MLI is superior to TNP-MLI [23]. Schweitzer et al. in [25] estimated the power loss and control performance of three-level TNP-MLI and underlined with a 10-kW hardware prototype. The authors in [26] dealt with TNP-MLI for PV application with two separate MPPTs for the DC-Link and investigated conduction loss for the inverter. Some similar structure of T type MLI with different PWM scheme is also discussed in [27–30]. The NPC-MLI majorly suffered by DC-link capacitors balancing. There are numerous papers published in literature. A few concentrate on the self-balancing idea with capacitors voltage switching ripple reduction as well. The self-balancing method is investigated in three-level NPC-MLI using carrier-based modulation [31–38]. The same self-balancing method nearest three vector selection is used in Hybrid Neutral Point Clamped transformerless inverter [39,40]. In [41], a modified SVM method is suggested with over modulation operation using small, medium, and zero vector switching vectors for mitigating the zero-sequence circulating current. In [42], the new CMV-eliminated SVM coordinate system is proposed. This method uses simple mathematical calculation to locate the reference vector and its nearest three CMV vectors duty cycles. However, in this method the inverter maximum output voltage is restricted

up to $V_{DC}/2$. Similarly, in [43] a method is investigated to yield output voltage more than $V_{DC}/3$, the amplitude of CMV could reduce only until 33% from the total CMV. Most importantly these CMV elimination methods affect the DC-link capacitors charging and discharge characteristics, which cause the large harmonics in the inverter output voltage and current waveforms. Current controller is the foremost request for grid-connected inverters. There are varieties of current controller discussed in the literature [32]. Based on the aforementioned discussion the PV-grid-connected TL NPC inverters should have improved inverter performance without any shoot-through burden on the inverter legs, commendable current controller, proper PWM to take care of the harmonics, switching frequency common-mode current (CMC), and leakage current.

The proposed TL-TNC-MLI has a simple structure with a PV-connected grid system using suitable hysteresis current control method to control the split inductor current of the grid. The hysteresis current control strategy directly measures the error current and functioning of the inverter switching pulses regarding PV output and grid situations. The systems proposed in this paper are aimed at proposing the TNP-MLI concept in single-phase transformer-less grid-connected system without shoot through and no switching frequency common-mode voltage issue. MATLAB/Simulink simulations confirm the proposed arrangements and experimentations of FPGA-based 1.5 kW PV tied TL-TNP-MLI grid-connected setup.

## 2. Analysis of Transformer-Less Multilevel NPC-MLI

### 2.1. Single-Phase Three-Level Transformer-Less Multilevel NPC-MLI

In general, a PV-based inverter functions under a considerably lower power range than the rated power. Besides, as conversed in the introduction, the NPC-MLI schemes seem more prominent since the IGBTs and diodes have comparatively larger state voltage/current ratio at the range of low and medium powers. Therefore, TNP-MLI is adequate for PV applications and power circuitry as shown in Figure 1a. Here, the clamping diodes ($D_1$–$D_2$) are absent, and it contains only 4-semiconductors (4-IGBTs: ($S_1$–$S_4$), 4-anti-parallel diodes (FWD: $D_{a1}$–$D_{c4}$), and two dc-link capacitors ($C_1$ and $C_2$). Considering medium power TNP-MLI topology, top and bottom switches ($S_1/D_1$ and $S_4/D_4$) work with 1200 V, and middle switches ($S_2/D_2$ and $S_3/D_3$) operate only at 600 V (half of the top and top and bottom switch voltage range). Unlike three-level DC-MLI, no series association of devices has to block the whole dc-link voltage $V_{dc}$ [24]. In the proposed TNP-MLI transformer-less topology the mid-point (C) of two DC-link/neutral point capacitors ($V_{C1}$ and $V_{C2}$) connects to the PV cluster mid-point and grid neutral (G). Table 1 shows the commutation interval, voltage and current path for TNP-MLI for inverting and rectifying modes.

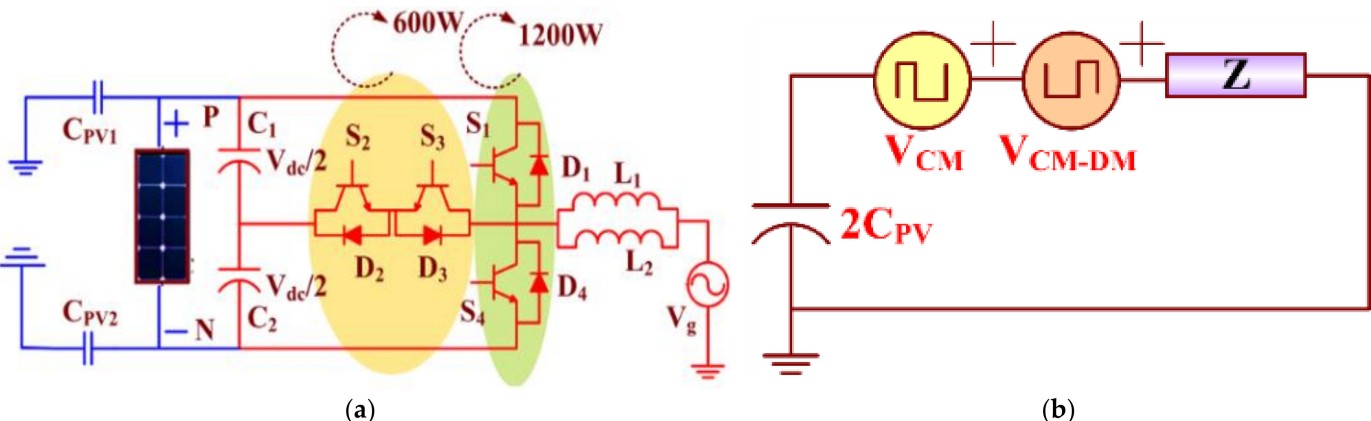

**Figure 1.** Power Circuit (**a**) Power circuit of single-phase transformer-less TNP-MLI. (**b**) Common-mode voltage model equivalent circuit single-phase transformer-less TNP-MLI.

**Table 1.** Commutation interval, voltage, and current path for transformer-less TNP-MLI.

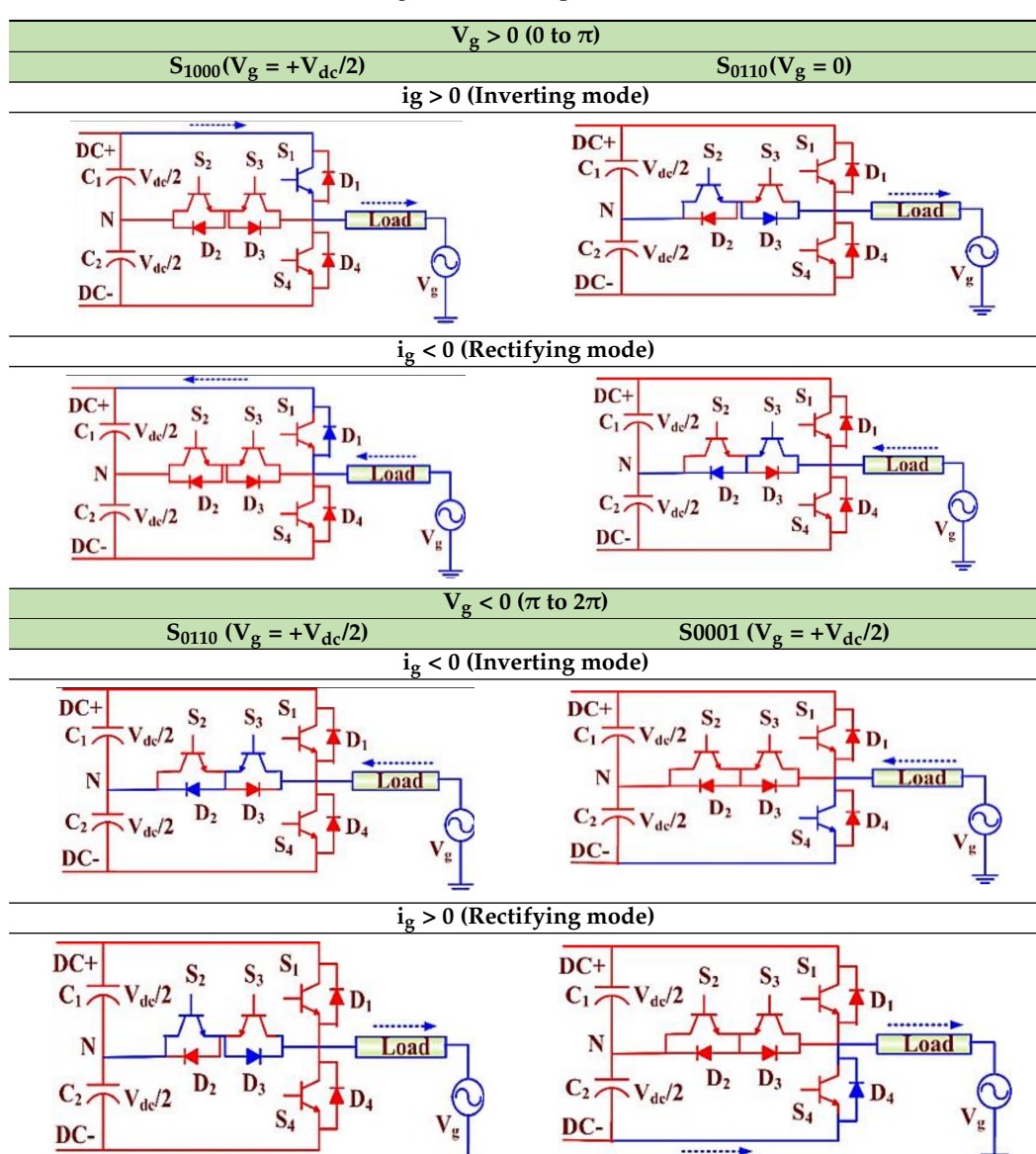

### 2.2. Single Modes of Operation in PV-Tied Single-Phase Three-Level Transformer-Less TNP-MLI Topology

The inverter switching frequency CMV of transformer-less PV inverter remains the birthplace of the leakage current coming from the inverter and the load ground [14,15]. In this system, there is a parasitic capacitance of the PV array (CG-PV) associated between common-ground and PV. However, in TL inverters, this switching frequency CMV behavior is significantly present, which leads to leakage ground currents on the inverter [4,10]. Hence, the inverter switching operation or topology structure must be investigated with respect to high-frequency CMV.

The proposed TL-TNP-MLI topology investigates the simplest CMV, and DMV based on the paper [15,26] and the PV-fed TL-TNP-MLI equivalent circuit illustrated in Figure 1b. Here, the DMV ($V_{DM}$) to CMV ($V_{CM}$) influence by $V_{CM\_DM}$, is equal to $-V_{DM}/2$.

During a positive half-cycle,

$$V_{CM\_DM} = \frac{V_{DM}}{2} \tag{1}$$

$$V_{CM\_DM} = \frac{V_{AN} + V_{CN}}{2} \tag{2}$$

$$V_{DM} = V_{AN} - V_{CN} \tag{3}$$

During a negative half-cycle,

$$V_{CM} = \frac{V_{BN} + V_{CN}}{2} \tag{4}$$

$$V_{DM} = V_{BN} - V_{CN} \tag{5}$$

The total CMV, including $V_{CM\_DM}$

$$V_{TCM} = V_{CM} + V_{CM\_DM} \tag{6}$$

The various possible switching operations of PV-tied TL- TNP-MLI are investigated for positive as well as negative half-cycle of grid voltage, as displayed in Figure 2a–d. Before going to the modes of operation, the following assumptions are made: (1) all IGBTs ($S_1$–$S_4$) are considered as ideal devices; (2) DC-link capacitors ($C_1 = C_2 = V_{PV}/2$) equally divide the DC-link voltages; and (3) the inverter operates at unity power factor (PF), i.e., inverter current $i_L$ and grid voltage ($V_g$) are in zero degrees phase shift.

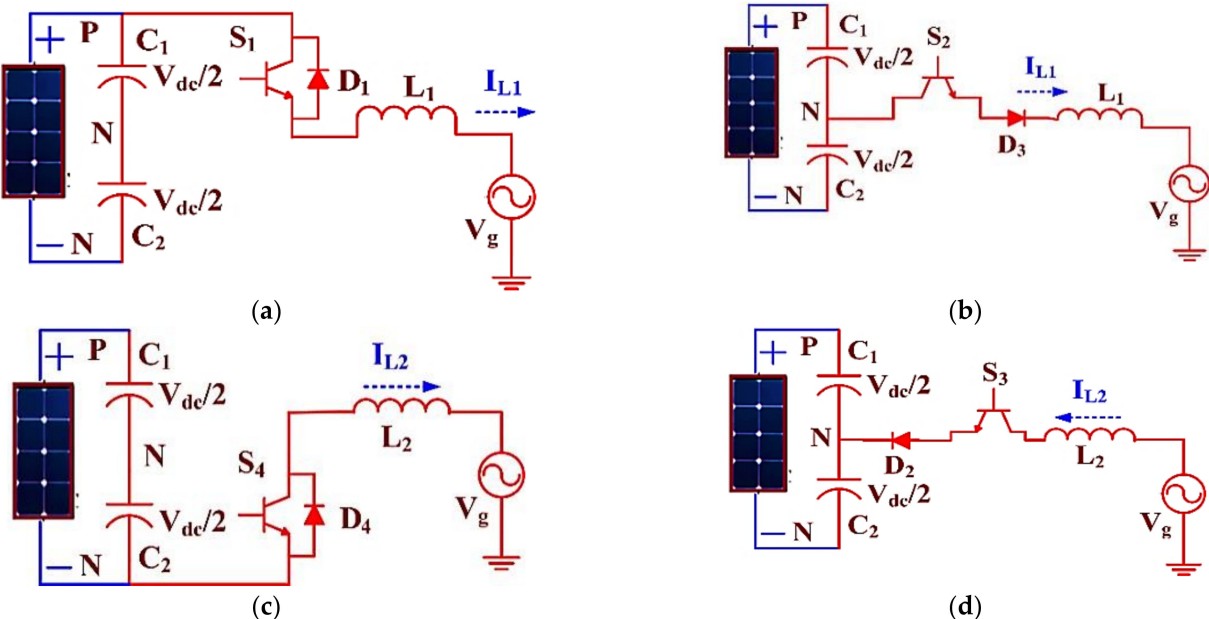

**Figure 2.** Mode of operation of transformer-less TNP-MLI: (**a**). Mode 1, (**b**). Mode 2, (**c**). Mode 3, (**d**). Mode 4.

Mode 1 ($S_1$ is ON, and $S_2$, $S_3$, and $S_4$ are OFF): Now voltage across the capacitor $C_1$ ($V_{C1}$), i.e., $V_{CA} = (1/2) V_{PV}$ supply to the grid via $L_1$. During this period, inductor $L_1$ current ($i_{L1}$) increases to $L_1 \frac{di_{L1}}{dt}$

$$\left. \begin{array}{l} L_1 \frac{di_{L1}}{dt} = \frac{1}{2}V_{PV} - V_g \\ V_{AM} = V_{PV}; V_{BM} = V_g - \frac{1}{2}V_{PV}; V_{CN} = \frac{1}{2}V_{PV} \\ V_{DM} = \frac{1}{2}V_{PV}; V_{CM} = \frac{3}{4}V_{PV}; \\ V_{CM\_DM} = \frac{1}{4}V_{PV}; V_{TCM} = \frac{1}{4}V_{PV}; \end{array} \right\} \tag{7}$$

Mode 2 ($S_2$ is ON, and $S_1$, $S_3$, $S_4$ are OFF): During this mode, zero potential points connected to the grid and output voltage of the bridge leg are zero, i.e., $V_{CA} = 0$. During this period, $i_{L1}$ starts to decrease to zero.

$$\left. \begin{array}{l} L_1 \frac{di_{L1}}{dt} = 0 - V_g \\ V_{AM} = \frac{1}{2}V_{PV}; V_{BM} = V_g; V_{CN} = \frac{1}{2}V_{PV} \\ V_{DM} = 0; V_{CM} = \frac{1}{2}V_{PV} \\ V_{CM\_DM} = 0; V_{TCM} = \frac{1}{2}V_{PV} \end{array} \right\} \tag{8}$$

Mode 3 ($S_4$ is ON, and $S_1$, $S_2$ and $S_3$ are OFF): Now voltage across the capacitor $C_2$ ($V_{C2}$), i.e., $V_{CB} = (1/2)V_{PV}$ supply to the grid via $L_2$. At this period, inductor $L_2$ current ($i_{L2}$) increases to $L_2 \frac{di_{L2}}{dt}$.

$$\left. \begin{array}{l} L_1 \frac{di_{L1}}{dt} = -\frac{1}{2}V_{PV} - V_g \\ V_{AM} = -V_g - \frac{1}{2}V_{PV}; V_{BM} = 0; V_{CN} = \frac{1}{2}V_{PV} \\ V_{DM} = -\frac{1}{2}V_{PV}; V_{CM} = \frac{1}{4}V_{PV} \\ V_{CM\_DM} = \frac{1}{4}V_{PV}; V_{TCM} = \frac{1}{2}V_{PV} \end{array} \right\} \tag{9}$$

Mode 4 ($S_3$ is ON, and $S_1$, $S_2$, $S_4$ are OFF): During this mode, the zero potential points connected to the grid and output voltage of bridge leg is zero, i.e., $V_{CB} = 0$. During this period, $i_{L2}$ starts to decrease to zero.

$$\left. \begin{array}{l} L_1 \frac{di_{L1}}{dt} = 0 - V_g \\ V_{AM} = -V_g; V_{BM} = \frac{1}{2}V_{PV}; V_{CN} = \frac{1}{2}V_{PV} \\ V_{DM} = 0; V_{CM} = \frac{1}{2}V_{PV} \\ V_{CM\_DM} = 0; V_{TCM} = \frac{1}{2}V_{PV} \end{array} \right\} \tag{10}$$

During the commutation modes, unlike NPC-MLI, "short" or "long" commutation paths are not present in TNP-MLI topology; all paths are having the same geometric length and have one outer switch (indices 1 or 4; either IGBT or diode) and two inner switches (either $T_2$ and $D_3$ or $T_3$ and $D_2$). Table 2 shows the commutation interval voltage and the current path for the four operating modes (positive voltage and positive current, a positive voltage and negative current, negative voltage and negative current, and negative voltage and positive current).

The TL-TNP-MLI providing uniform voltage transition (zero to $+V_{PV}/2$ and zero to zero $-V_{PV}/2$) according to positive as well as negative half-cycle of grid voltage, which diminishes current ripple of inductor naturally, and hence TL-TNP-MLI requires less inductance range compared to the dual-buck half-bridge inverter [16]. Besides, based on the CMV calculation on the different mode of the inverter, it could be understood that the proposed transformer-less TNP-MLI produces constant VTCM, and it will not have any leakage current. This benefit comes about from the MLI topology structure naturally produces less CMV compared to other inverters [24].

The proposed TL-TNP-MLI topology compared with conventional ones is presented in the Table 2. Inverter efficiency performance is considered with different switching frequency ranges from 2 kHz to 25 kHz. The efficiency of the TL-TNP-MLI is stupendous under medium switching frequencies from 3 kHz to 20 kHz. Here, the significant improvement of the T type NPC topology is reduction in their switching losses as the commutation voltage of the 1200-V IGBTs is only 350 V instead of 700 V. When switching frequency is above 20 kHz, the NPC-MLI topology gave a better performance. Noteworthy, the low-voltage application is like the PV-grid-connected system; the TL-TNP-MLI is the best choice concerning cost and efficiency.

**Table 2.** Commutation interval, voltage, and current path for four operating modes.

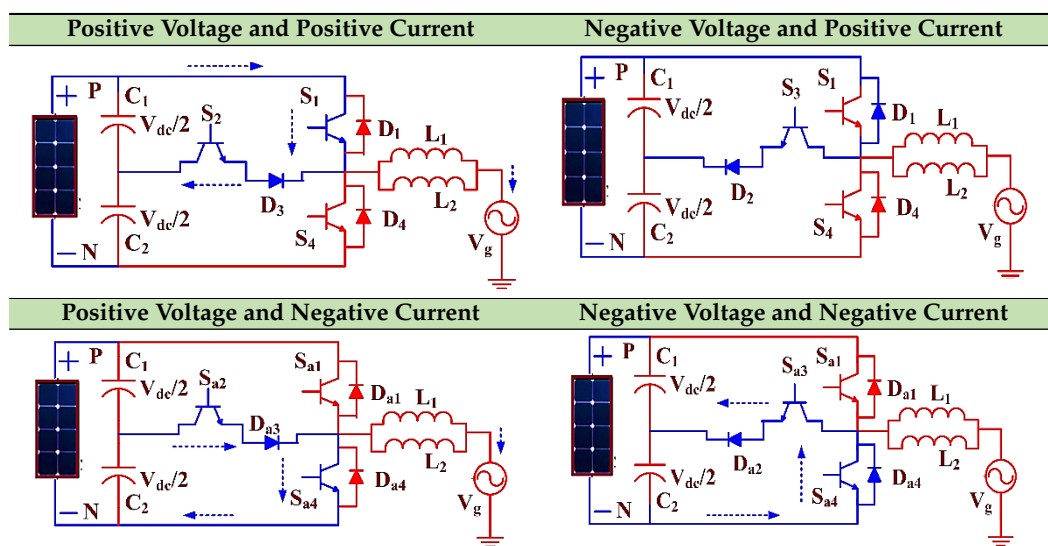

To summarize, in both I-Type and T- Type NPC MLIs, the zero vector's conduction loss leads to the entire conduction losses besides even the complete semiconductor power losses. Nevertheless, the T- Type NPC MLI only improves the T- Type NPC MLI non-zero vector conduction losses, as well as the current path of T-type zero vectors is the same as that of I-type. Thus, the T-type improvement over I-type is considerably restricted due to the unimproved zero-vector conduction loss.

Diode Free NPC-MLI, four CoolMOSFETs replace the IGBT and diode middle bidirectional switch. In the proposed topology, there is no diode involved in the current path supposing the unity power factor. In all the compared topologies the inner and outer power switches need 1200 V operation voltage, however the proposed one needs only 600 V in both inner and outer power switches. Unlike the conventional MLIs, Shoot through Short circuit problems is not there in the proposed one.

### 3. Control System of PV Grid Tired Connected Transformer-Less TNP-MLI

The controlling of the inductor current, and improved current controller (CC) to take care of the inductor charging and discharging nature to avoid the shoot through problem in inverter bridge legs are much appreciated for this proposed TL-TNP-MLI.

#### 3.1. Design

The hysteresis current controls (HCCs) have precise tracking competence to track the inverter current with the grid voltage to ensure unity PF and current waveform quality [31]. Figure 3a displays the structure block diagram of HCC. Here, the reference current ($i_{ref}$) which is associated with actual current $i_{act}$ creates the error current $(\overrightarrow{\varepsilon_i}) = (iL_{ref} - iL_{act})$, and yields the control input u (t) among two hysteresis bands: upper hysteresis bands (UU) and lower hysteresis bands (BL) [28–31]. Such control input u(t) drives the PWM block to give the perfect interface between inverter and grid. The challenge here is to maintain the $i_{act}$ close to the $i_{ref}$ inside the hysteresis boundaries [9]. To ensure that, the proposed grid-connected TL-TNP-MLI interface is investigated with HCC to verify zero $i_L$ (inverter current) sooner than the zero-crossing of grid voltage "$V_g$". Figure 3b illustrates the hysteresis switching assortment for the proposed HCC.

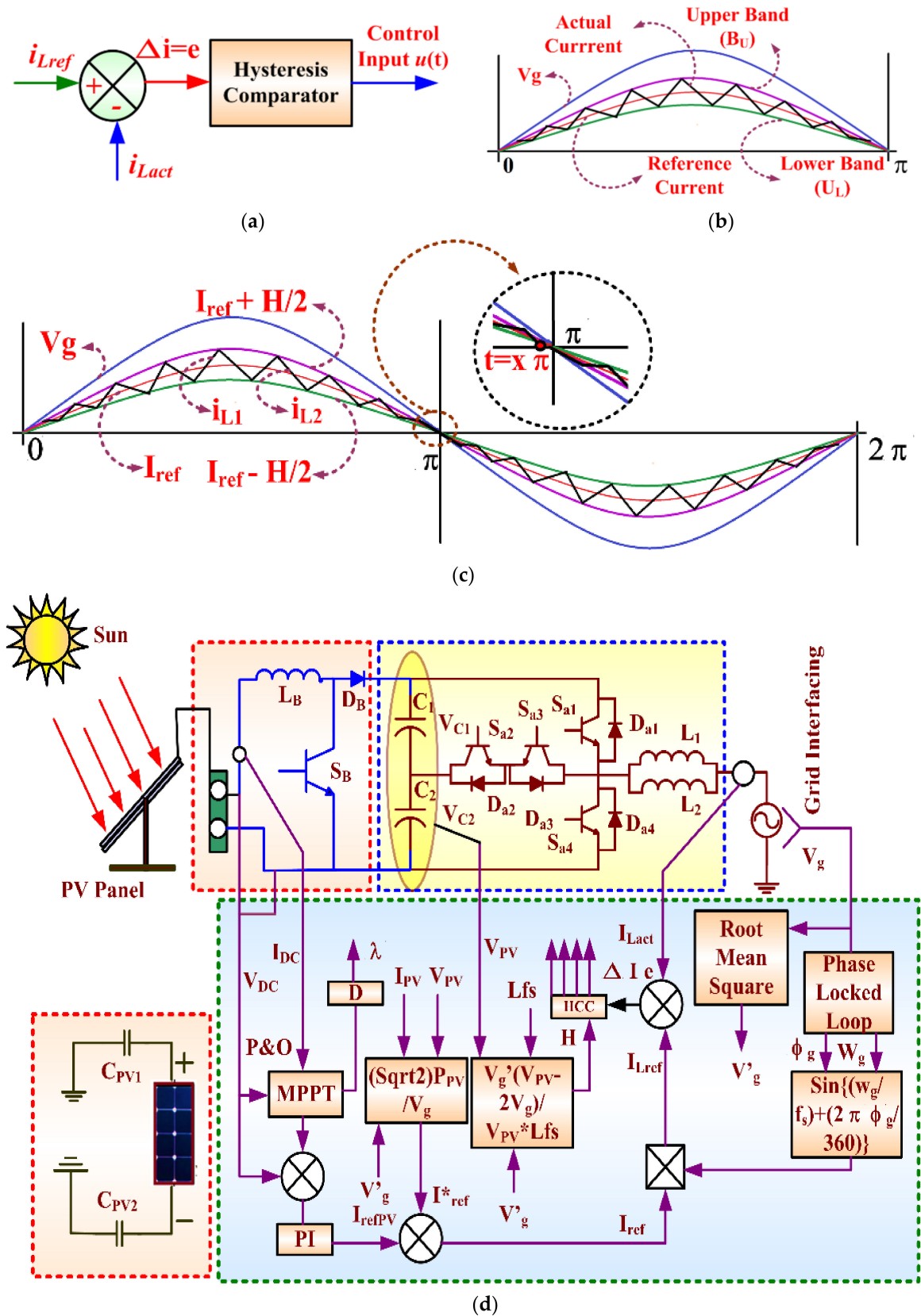

**Figure 3.** Control Logic (**a**) Basic block diagram. (**b**) Hysteresis switching assortment. (**c**) HCC band and inductor current control with grid voltage. (**d**) Closed-loop PV connected single-phase three-level transformer-less TNP-MLI PV-grid-connected system control blocks.

According to Figure 3c, the x point has been chosen just before π (x should be less than 180°). The interval between x to xπ is chosen more than the switching frequency ($f_s$) to avoid the pulse dropping.

$$\frac{\int_{x\pi}^{\pi} [V_g\sin(wt) + V_S(ON)]}{L} \geq i_{ref}\sin(x\pi) + \frac{H}{2} \tag{11}$$

Here, $V_S(ON)$ = ON-state voltages of power switch $i_{ref}$ is current grid reference, H = bandwidth of HCC, and L = filter inductor. Fixed frequency HCC is achieved by computing hysteresis bandwidth H from temporary value of the $V_g$ and variable PV array output voltage VPV. Hence the "H" is calculated based on the ripple of the inductor current, the transient value of the grid voltage $V_g$, filter inductor L, switching frequency $f_s$, and the output voltage of PV array $V_{PV}$. The fixed frequency bandwidth of HCC,

$$H = \frac{V_g(V_{PV} - 2V_g)}{V_{PV}Lf_s} \tag{12}$$

Here the filter has been chosen based on the inductor value since the filter design is highly difficult for high stability, and fast dynamic response nonlinear controller HCC. Few researchers have investigated on hysteresis band calculation including filter inductance [33]; among them the real-time hysteresis regulator design built on the switching frequency is the successful technique to estimate the H value.

### 3.2. Closed-Loop PV Tie Grid-Connected Control Strategy of Proposed Transformer-Less TNP-MLI

The suggested grid-connected PV connected TL-TNP-MLI control blocks representation is shown in Figure 3d. Here, the system is classified into four modules, namely; PV module, boost converter, TL-TNP-MLI, and grid interfacing control system. In the solar module, the measurement of PV voltage and current is given to P&O technique centered MPPT.

The PLL is implemented on the grid side to calculate the grid angle $\theta_g$ and utility grid nominal frequency ($w_g = 2\pi f_o$), $f_o$ is grid voltage frequency for the grid synchronization. Here, the most straightforward zero-crossing detection (ZCD) algorithm has been used. The grid voltage RMS ($V_g'$) is measured and given to the $I_{ref}^*$ estimated block, where $I_{ref}^*$ sampling from $V_{PV}$, $I_{PV}$, and RMS values of grid voltage $V_g'$ are as shown below,

$$I_{ref}^* = \frac{\sqrt{2V_{PV}}}{V_g'} \tag{13}$$

The $I_{ref}^*$ is compared with PV reference current ($I_{PV\text{-}ref}$) and estimating the final grid reference current dc- component ($I_{ref}$), which is multiplied with sin (($w_g + f_s$) + ($2\pi \theta_g/360$)) to calculate the final grid reference current ($I_{ref}$). Here, this $I_{ref}$ is also called an inductor reference current ($I_{Lref}$). The actual inductor $I_{Lact}$ and $I_{Lref}$ gives the error of inductor current $\Delta_{Lie}$ based on the $\overrightarrow{\varepsilon_i} = (i_{Lref} - i_{Lact})$, finally, the calculated real-time hysteresis band H using (12) and $\Delta_{Lie}$ gives gate pulses for the inverter switch ($S_1$ to $S_4$).

### 4. Numerical Simulation Results

To verify the validity of the proposed grid-connected PV-fed TL-TNP-MLI method, the simulation model is customized using MATLAB/Simulink (Table 3 simulation parameters). The 1.5 kW PV array module was modeled with P&O MPPT algorithm, functioned at different irradiation and temperature. The Figure 4a,b shows the PV module simulation results, here the P&O MPPT controller tracks PV array MPP within the range of 1.5 times of the grid frequency. For simulation analysis of the proposed inverter with HCC, 240 V input voltage (dc-link) is maintained from PV-connected boost converter and it is adequate to inject current into grid, as the $V_g$ is taken as 230 V. Range of switching frequency is

from 5 kHz to 40 kHz. The main objective of the proposed grid interfacing is explained through the validation results of inductor current ($i_L$), and grid voltage ($V_g$) projected in the same scale as shown in Figure 4c. Here, $V_g$ is multiplied by 0.03 to bring in-shape the grid voltage $V_g$ with load current $I_L$ to view both on the same scale. From the waveform, it is understood that $V_g$ and $I_L$ are meeting each other in the same zero-crossing point and ensuring the grid synchronization. By using the zoomed view on the zero-crossing point in Figure 4d, the HCC prevents the high-frequency signal just before the $V_g$ reaching zero scales, as shown in Figure 4d. Figure 5a shows the inductor current ($i_{L1}$ and $i_{L2}$) and Figure 5b illustrates the current tracking between actual inductor $I_{Lact}$ (inverter current) and grid current reference $I_{Lref}$.

**Table 3.** Simulation parameter of PV and grid-connected TL-TNP-MLI.

| Parameters | Values |
|---|---|
| PV module design/W | 1250 W |
| Grid voltage/V and grid frequency/Hz | 240 V/50 Hz |
| Inverter power rating/W | 1000 W |
| dc-link capacitors (C1 and C2)/µF | 470 µF/500 V |
| Filter inductors (L1 and L2)/mH | 4 mH |
| CM Capacitance (CY1 and CY2)/nF | 2.2 nF |
| Parasitic Capacitance (CPV1 and CPV2)/µF | 0.2 µF |

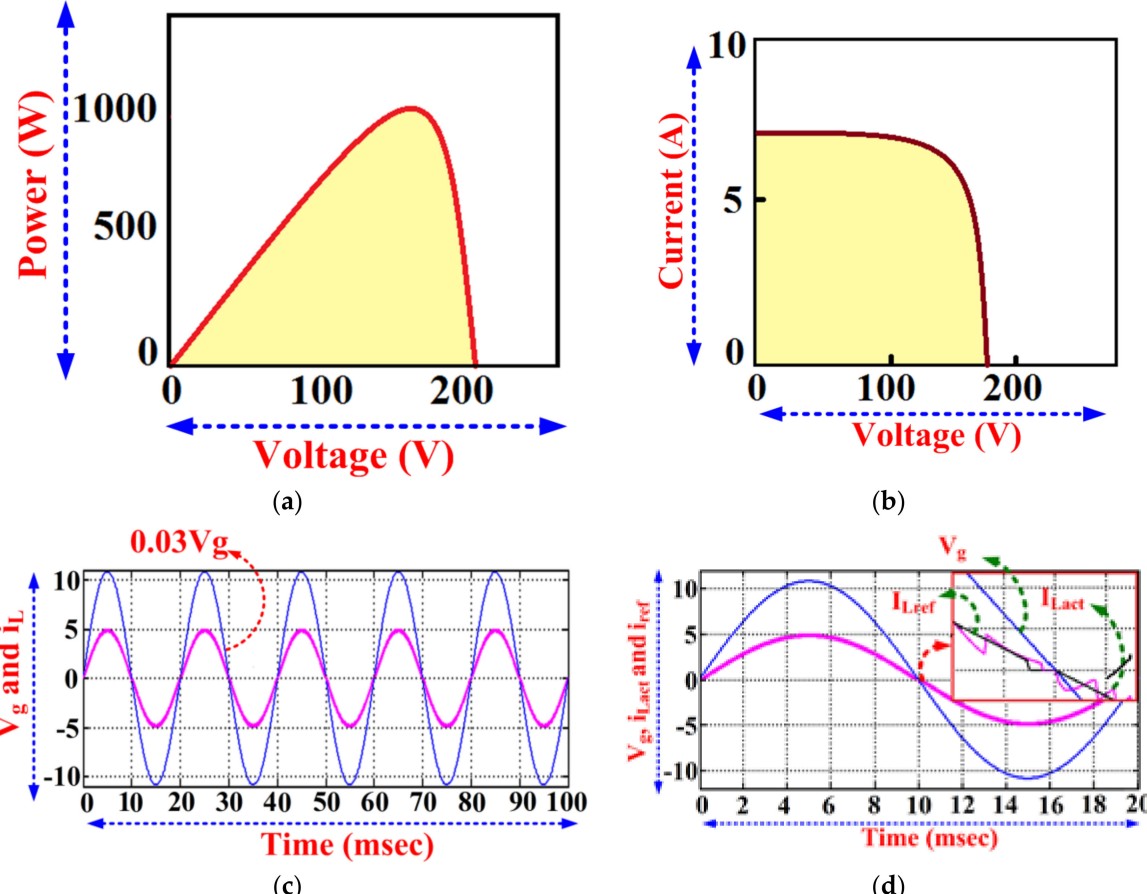

**Figure 4.** PV characteristics and the simulation results of the grid -PV connected TL-TNP-MLI (**a**) P-V, (**b**) I-V, (**c**) $i_L$ and grid voltage $V_g$, (**d**) zoomed view on $i_L$ verse $V_g$.

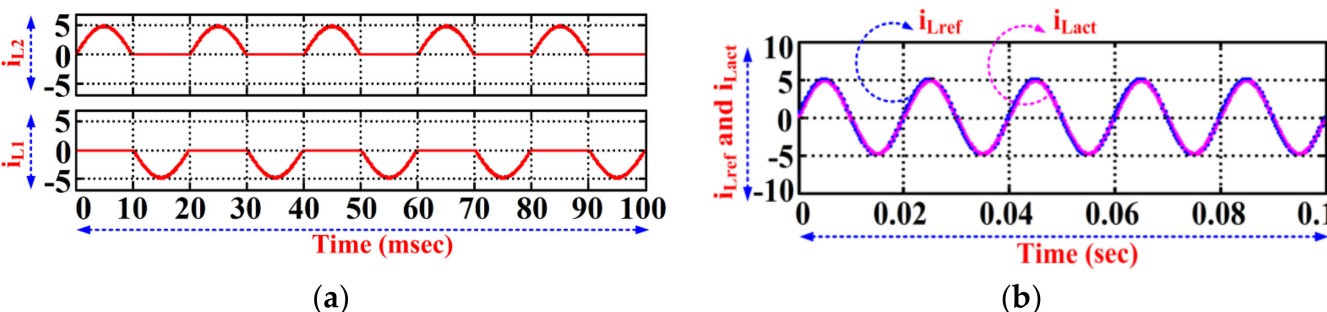

**Figure 5.** Simulation results. (**a**) Inductor current ($i_{L1}$ and $i_{L2}$). (**b**) $I_{Lact}$ (inverter current) tracks against $I_{Lref}$.

Figure 6a,b shows the waveforms of inverter current and grid voltage with harmonic spectrum of the inverter current single-phase PV-fed grid-connected transformer-less TNP-MLI operating under fixed and variable hysteresis band H. Here, the spectrums are worked out till 500th order (25 kHz/50 Hz); it can be seen the THD percentage for both fixed and variable band are less as 5% and 8% respectively.

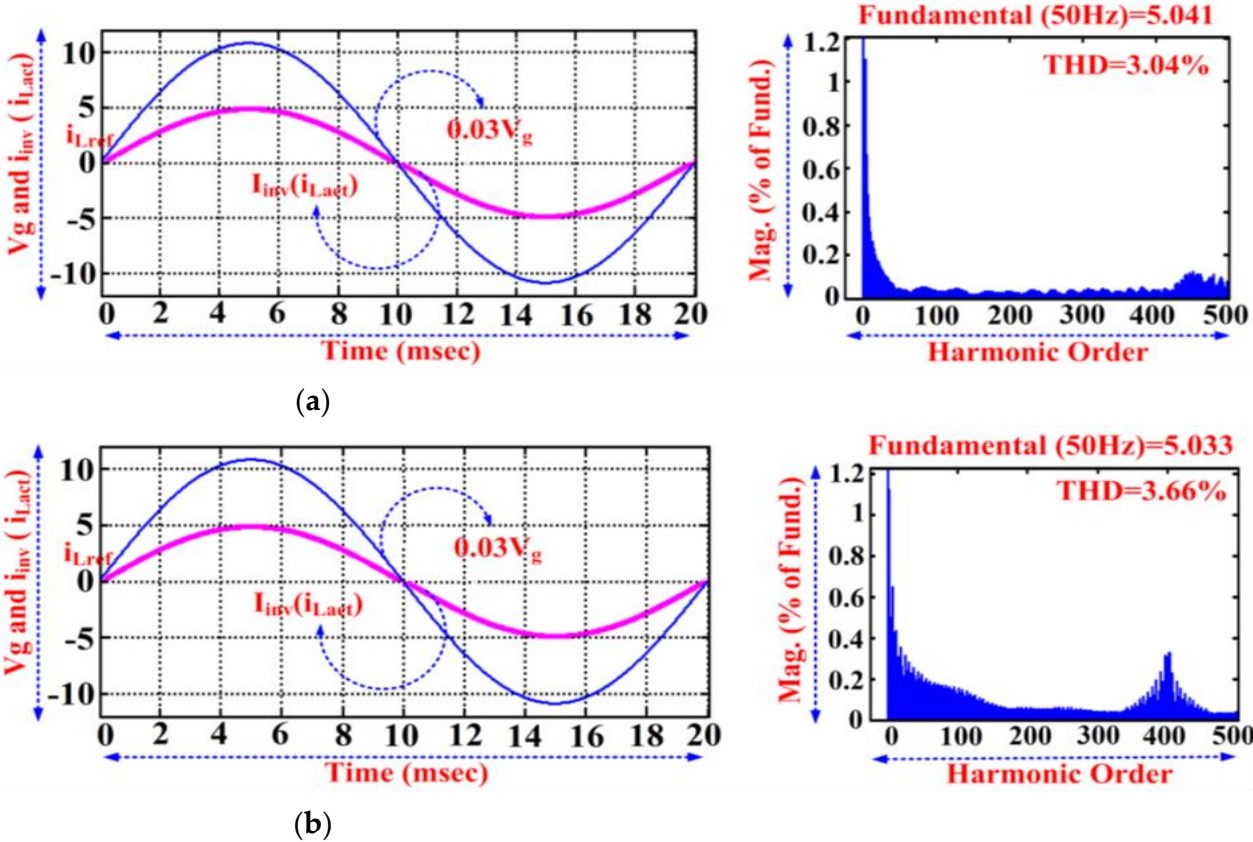

**Figure 6.** Hysteresis current control performance on the proposed PV tied inverter (**a**) fixed hysteresis band response on $i_L$, $V_g$, and energy spectrum of grid current and corresponding harmonics spectrum. (**b**) Variable hysteresis band response on $i_L$, $V_g$ and energy spectrum of the grid current and corresponding harmonics spectrum.

The inductor current accomplishes the ability of tracking with high-precision of the given current reference concerning PV and grid variations. According to grid interfacing international standards (UL1741 and NEC690), inverter and its controllers must be capable of connecting to the grid with all protections. To ensure this decisive factor, the proposed grid interface is tested with different variation in PV and grid, as shown in Figure 7. From Figure 7a–c, it could be understood that, once the inverter switching pulses are turned OFF, then the inverter import to the grid stopped. Next, the study is performed for the dynamic

conditions such as a sudden upsurge and fall on PV input voltage, grid voltage, and grid-current reference, as shown in Figure 8a–c, respectively. In Figure 8c the grid current drops from 3.5 to 1.8 msec and at 6.8 msec, the actual current touches the reference current $i_{ref}$ using constant frequency variable hysteresis band H for 1/5 of the grid frequency interval itself. Figure 8c also displays the closest view of inverter current and reference current while there is a sudden variation in the grid current. In the same way when the $V_g$ and PV input voltage change suddenly (see Figure 8a,b), the inverter interfacing algorithms on HCC respond rapidly to get the actual load current match the reference current.

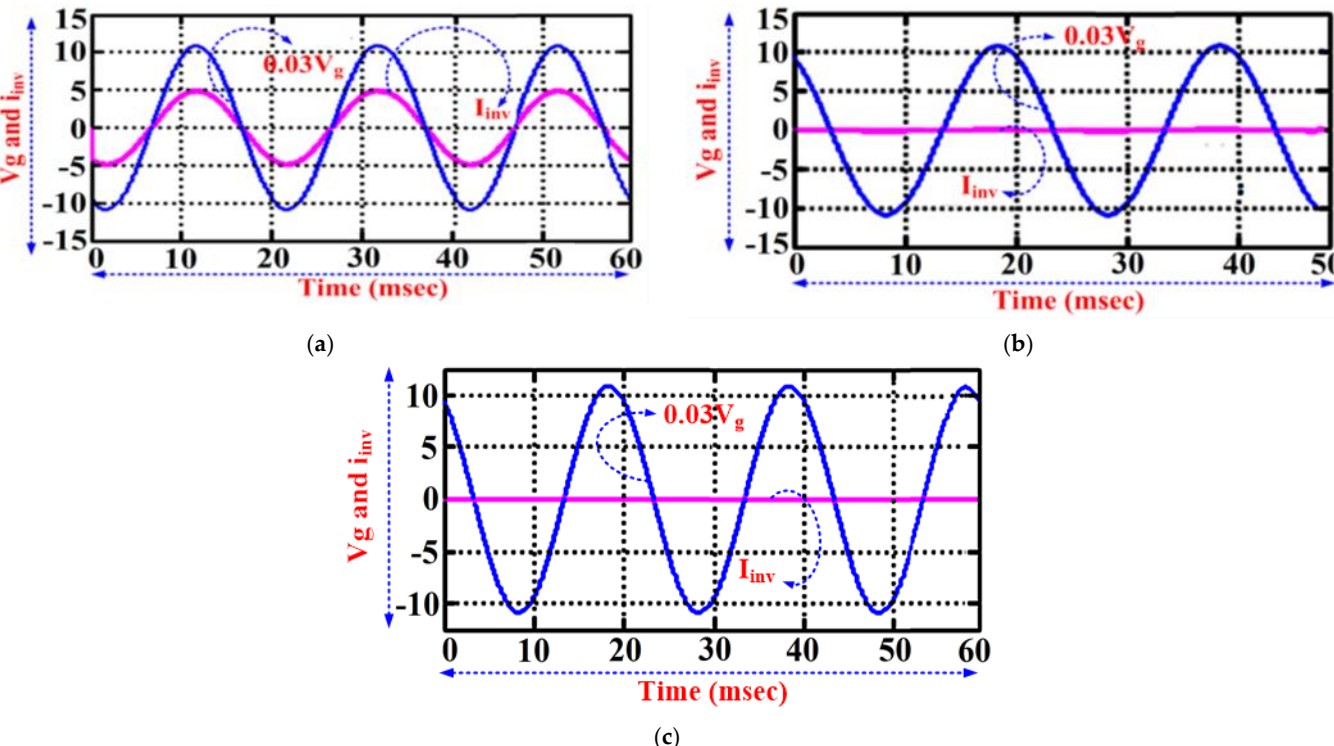

**Figure 7.** Steady operation under various grid-current for HCC at variable constant-frequency band. (**a**) At rated grid current, (**b**) 5% of the rated grid current, (**c**) zero grid current.

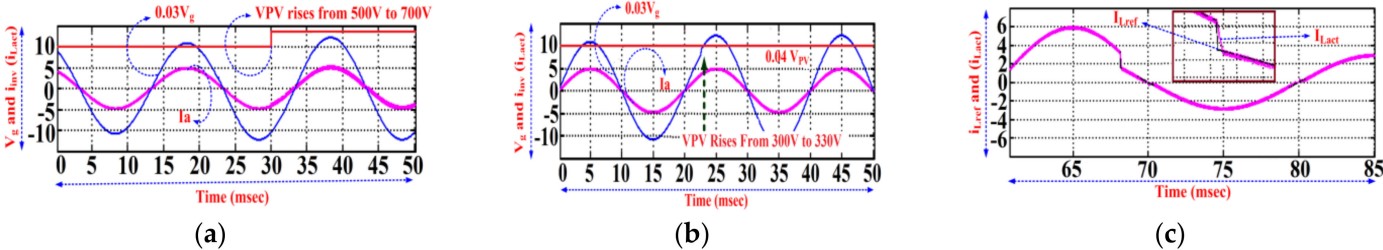

**Figure 8.** The performance of the PV-connected grid-connected inverter in different grid variations. (**a**) Change in PV voltage. (**b**) Change in grid voltage. (**c**) Grid current reference.

Finally, the common-mode current (leakage current, $i_{leakage}$) is measured between PV ground and grid common ground, as shown in Figure 9a. The total CMV (TCM)) produced by the TL-TNP-MLI is a constant ($P_{PV}/2$) for all the modes of operation. Hence leakage current observed here is very low (also almost zero). In this section, the TL grid-connected fitness of the proposed TL-TNP-MLI is presented in comparison with the other reported inverters such as DB-HBI, NPC-MLI, and TL-NPC-MLI. As shown in Table 4 the comparison considers various aspects of TL inverter requirements like the number of power devices (IGBT, FWD, and clamping diodes), voltage and current stress on the power switches, switching device power factor (SDPF), shoot-through issues, and leakage-current

phenomena [19,34,35]. Based on Table 4, it could be realized that the DB-HBI and NPC-MLI are in pathetic circumstances at the shoot-through problem, whereas the TL-NPC-MLI and TL-TNP-MLI are well-positioned except the filter inductors.

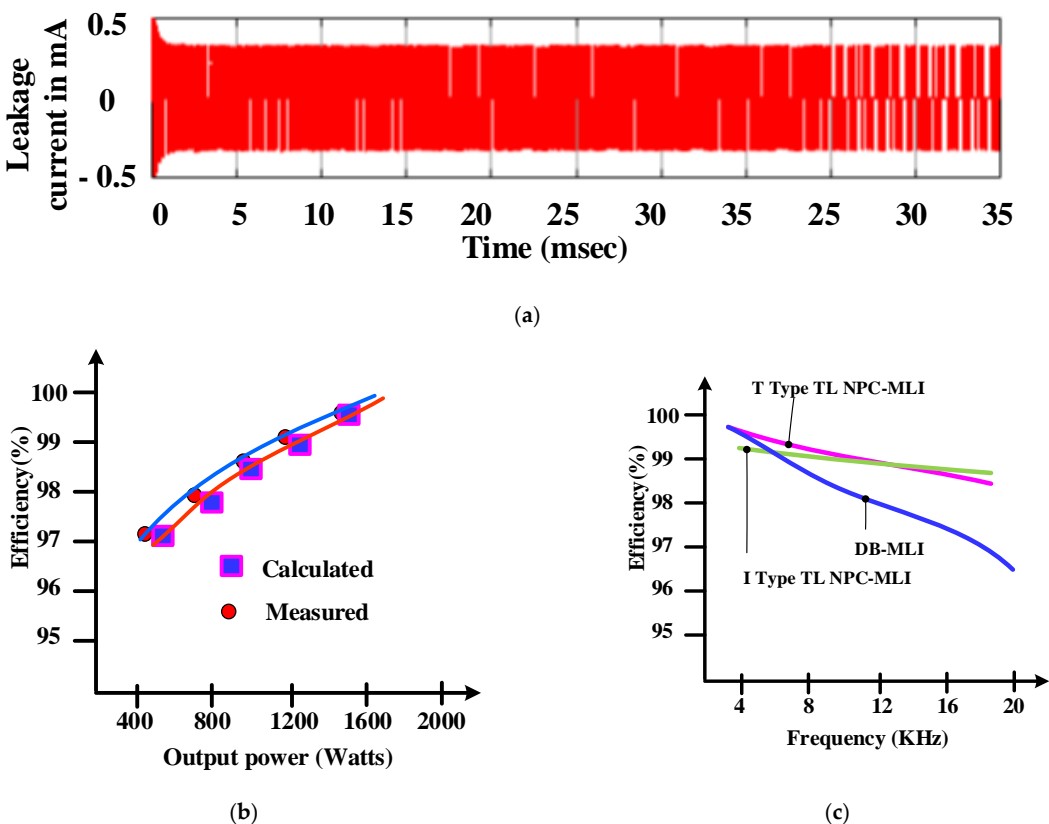

(a)

(b)                                                                 (c)

**Figure 9.** Simulation results. (**a**) Common-mode, $i_{leakage}$ in between inverter and PV module. (**b**) Calculated and measured efficiency of the TNP-MLI prototype. (**c**) Calculated efficiency comparison among TNP-MLI, NPC-MLI, and DB-HBI for Power = 1.5 kVA, $V_{peak}$ = 230 V, $I_{peak}$ = 5 A.

**Table 4.** Comparison of proposed inverter topology with the conventional ones.

| Inverter Topology. | No. of Switch/Leg | No. Of Clamping Diode/Leg | Top and Bottom Switch Voltage Rating | Inner Switch Voltage Rating | Short Circuit Problems |
|---|---|---|---|---|---|
| I-Type NPC MLI [13] | 4 | 2 | 1200 V | 1200 V | Possible |
| T-Type NPC MLI [19] | 4 | 0 | 1200 V | 600 V | Possible |
| Diode Free NPC-MLI [33] | 6 | 0 | 1200 V | 1200 V | Possible |
| DBHBI-MLI [11] | 4 | 2 | 1200 V | 1200 V | No |
| Proposed | 4 | 0 | 600 V | 600 V | No |

The proposed TL-TNP-MLI is entirely prevailing over the DB-HBI and NPC-MLI. When comparing TL-NPC-MLI with TL-TPC-MLI, device count, and selection of switch rating is much lesser (clamping diodes are absent), which results in less total SDPF. Hence, TL-TPC-MLI has the rewards of reduced switching and conduction loss, resulting in less SDPF compared to other topologies. Figure 9b illustrates the measured efficiency versus the output power of the TL-TNP-MLI, which shows better performance throughout the power ranges. The calculated efficiency is derived based on the power and a loss calculation formula in [23]. The power analyzer Yokogawa (power accuracy of ± 0.02%) is used for measurements of power for the fixed RL-load (R = 10Ω, L = 4 mH). Here, calculated efficiency is more than the measured one (due to the unaccounted losses on cabling and screw resistances). Further measurements exposed the other loss components produced by cabling resistance.

Next, the TL-TNP-MLI and other comparable inverter efficiency performance is considered with different switching frequency ranges from 2 kHz to 25 kHz and plotted in Figure 9c. The efficiency is stupendous under medium switching frequencies from 3 kHz to 20 kHz. Here, the significant improvement of the T type NPC topology is reduction in their switching losses as the commutation voltage of the 1200-V IGBTs is only 350 V instead of 700 V. When switching frequency is above 20 kHz, the NPC-MLI topology gave a better performance. Noteworthy the low-voltage application like PV-grid-connected system; the TL-TNP-MLI is the best choice concerning cost and efficiency. The proposed TL-TNP-MLI also having few limitations; (i) the maximum switching frequency is limited to 50 kW. Whereas the soft switching TL inverter can be operated up to 100 kHz. (ii) The outside side filter inductor must be same, otherwise large amount of DC compound will be produced by the inverter. (iii) The proposed TL-TNP-MLI is single grounded. Hence, high frequency DC oscillation may occur when the inverter operating at higher switching frequencies.

Three-level NPC-MLIs are widely used in motor control and PV-grid-connected applications compared to higher level MLIs. The NPC-MLI uses the number of switches and clamping diodes and provides the multiple voltage levels through the different phases. The DC-link capacitor connected in series and split the DC-link voltage equally to make the voltage levels in the output. The level increase in the NPC-MLI is constructed by increasing the DC-link capacitors, clamping diode, and increasing the switches. It is the main drawback of the NPC-MLI in maintaining the DC-link capacitor balancing. Hence the NPC-MLI topologies are limited to the three levels [8].

## 5. Hardware Implementation and Experimental Result

A 1.5-kW laboratory-scale prototype single-phase three-level TL-TNP-MLI is designed to verify the proposed PV-fed grid-connected system. Figure 10a displays the experimental setup of the proposed PV-fed grid-connected single-phase three-level TL-TNP-MLI. The existing designed three-phase TNP-MLI laboratory power module (Figure 10b) has been used for this proposed single-phase three-level TL-TNP-MLI by making use of their easy phase split options. Here, the bridge leg-1 switches ($S_1$, $S_2$, $S_3$, and $S_4$) are used for the proposed inverter (other switches are not used for the proposed work, they are totally isolated from supply and gate drivers). Here the outer switches (both the top and bottom switches ($S_1$ and $S_4$)) are Fairchild's FGA15N120ANTD type, and the inner switches ($S_2$ and $S_3$) are Mitsubishi Nch IGBTCT60AM-18F. The 330 μF/450 V DC-link capacitors ($C_1$ and $C_2$) are chosen to maintain the inverter DC-link voltage as 400 V.

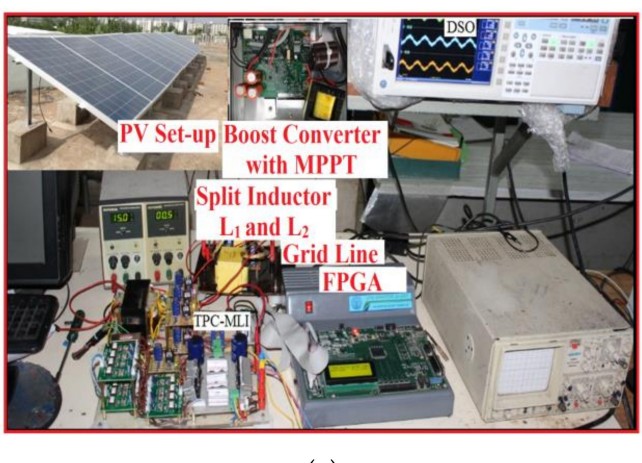

**(a)**

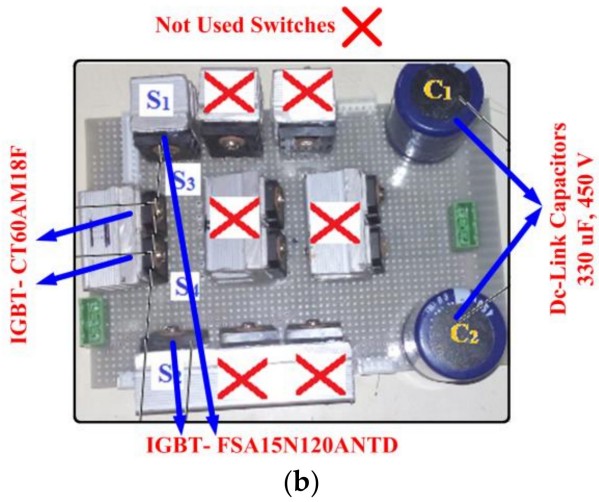

**(b)**

**Figure 10.** *Cont.*

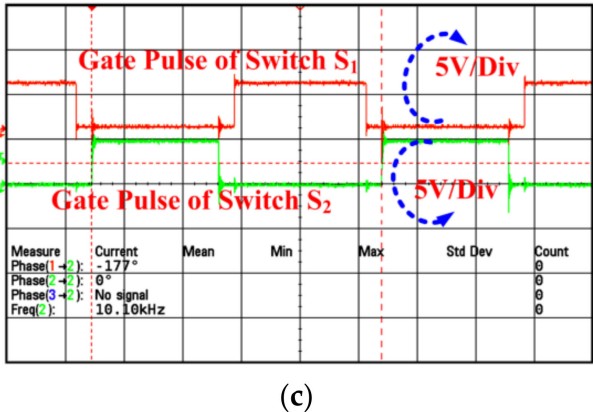

(c)

**Figure 10.** Experimentation Snapshot. (**a**) Proposed FPGA-based PV connected grid-connected transformer-less TNP-MLI. (**b**) Prototype of the TL-TNP-MLI. (**c**) The PWM pulses for top and bottom switches (S1 and S4).

The switching frequency ($f_s$) range is set from 3 to 25 kHz. At the nominal fs of 5 kHz, the inverter accomplishes a pure semiconductor efficiency of 99%. The additional power losses on the inverters are about 15 W, which are consumed by digital control and gate drive circuits. Initially, the inverter is tested with the PV system and boost converter for the constant DC-link voltage and when the HCC performance is confirmed with the prototype; a full-scale grid-connected transformer-less TNP-MLI was built and tested. All algorithms (MMPT, HCC, PWM, and grid interfacing) were implemented in a low-cost FPGA-Spartan-III processor through MATLAB/Simulink system generator. All the corresponding results were measured using a digital oscilloscope (Tektronix make) and power analyzer (Yokogawa make). Figure 10c indicates the inverter gate pulses of top and bottom switches ($S_1$ and $S_4$) via Tektronix DSO. All the inverter pulses are continually monitored and verified for the reliable operation of the inverter. Figure 11a shows the experimentation waveforms of inductor current ($i_{L1}$ and $i_{L2}$) of PV-tied grid-connected single-phase three-level TL-TNP-MLI. Here both the inductors ($L_1$ and $L_2$) are charging and discharging uniformly in opposite to one another. Figure 11b illustrates the inverter current ($i_L$) and its harmonic spectrum.

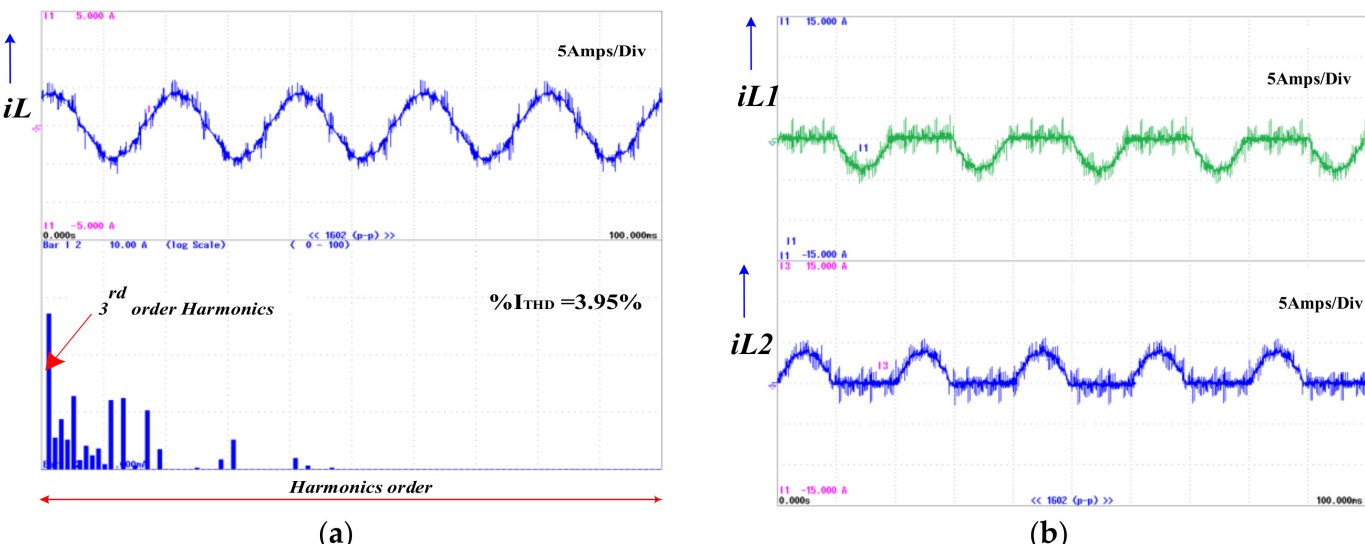

(a)　　　　　　　　　　　　　　　　(b)

**Figure 11.** *Cont.*

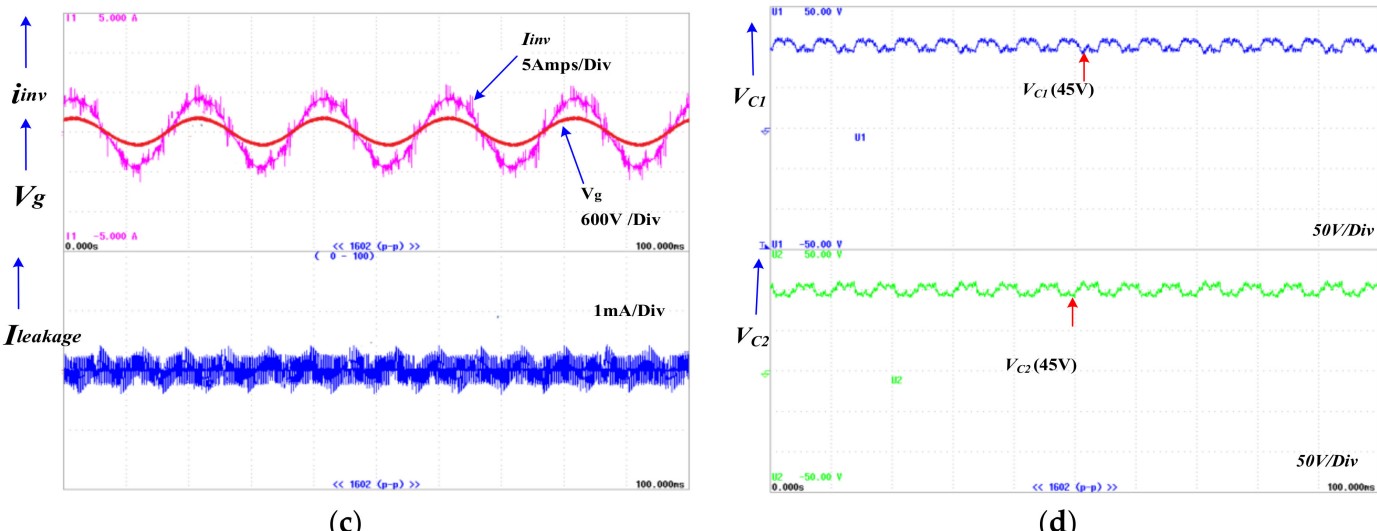

**Figure 11.** Experimentation results. (**a**) Inductor currents ($i_{L1}$ and $i_{L2}$). (**b**) TNP-MLI current and harmonics Spectrum. (**c**) $i_L$ and grid voltage $V_g$, besides leakage current measurements. (**d**) Waveforms of dc-link capacitors ($C_1$ and $C_2$).

The harmonic spectrum of $i_L$ is observed to be much similar to simulation value. Figure 11c shows inverter leakage current, grid voltage ($V_g$) and inverter current ($i_L$). To show the zero phase shifts in between $V_g$ and inverter current, the measurement of $V_g$ and $I_L$ is done using the same scope window as 600 V/div and 5 A/div respectively. On the basis of waveform, it is seen that the $V_g$ and $I_L$ are meeting each other in the similar zero-crossing point and ensure the inverter synchronization with grid. The leakage current of the proposed TL TNP-MLI is witnessed near to zero (20 mA). Figure 11d shows the two capacitor voltages ($V_{C1}$ and $V_{C2}$) the inverter DC-link. Besides, no much distortion is present in the inverter output current. The comparison of proposed inverter with existing inverter configuration. Is presented in Table 5. From the comparisons, it understands that the proposed TNP-MLI showing better choice for the TL inverter configurations.

**Table 5.** Comparison of proposed inverter with existing inverter configuration.

| Parameters. | Units/Rating | [16] | [11] | [19] | (Proposed) |
|---|---|---|---|---|---|
| | No. of switch | 2 | 4 | 4 | 4 |
| | $V_{DS}$(Peak)/V | 800 | 400 | 400 | 400 |
| Switches | $I_D$(rms)/A | 2.54 | 2.16/2.95 | 2.16/2.95 | 2.16/2.95 |
| | Switch SDPF/kVA | 4.009 | 4.008 | 4.008 | 4.008 |
| | Inner switch voltage rating | 1200 V | 1200 V | 1200 V | 600 V |
| | Top and Bottom switch voltage rating | 1200 V | 1200 V | 1200 V | 1200 V |
| | No of Diode | 2 | 2 | 2 | 0 |
| Clamping diodes | $V_{KA}$(peak)/V | 800 | 400 | 400 | - |
| | $I_A$(rms)/A | 0.47 | 0.93 | 0.94 | - |
| | Diode-SDPF/kVA | 0.752 | 0.744 | 0.744 | 0 |
| Total SDPF/kVA | SDPF/kVA | 4.816 | 4.832 | 4.832 | 4.008 |
| Inductor details | No. of inductor | 2 | 1 | 2 | 2 |
| | L value-L/mH | 16 | 4 | 4 | 4 |
| Short circuit problems ( Shoot through) | | Possible | Possible | No | No |
| Leakage current | | No | No | No | No |

## 6. Conclusions

This paper has presented an effective single-phase PV tied transformer-less three-level TNP-MLI grid-connected system designed as well as successfully implemented, with unique successfully implemented like assurances of no switching frequency. This TNP-MLI rallies better performance than NPC-MLI and another type of MLIs topologies, with its clamping diode-free circuit structure. The advantages of the proposed inverter with the PV tied TL application is compared with other inverter structures. Also, the proposed system confirms the healthier performance on steady-state and transient periods. MATLAB/Simulink simulations confirm the proposed arrangements and the experimentations of FPGA-based 1.5 kW PV-tied TL-TNP-MLI grid-connected setup.

**Author Contributions:** Conceptualization, P.M., C.B. and R.V.; methodology, P.M., C.B., L.M.-P. and R.V.; software, A.R., A.M. and R.U.; validation, P.M., C.B. and R.V.; formal analysis, P.M., C.B. and R.V.; investigation, P.M., C.B. and R.V.; resources, J.L.M., R.V.; data curation, P.M., C.B. and R.V.; writing—original draft preparation, P.M., C.B. and R.V.; writing—review and editing, J.L.M.; supervision, R.V., C.B. and L.M.-P.; project administration, R,V.; funding acquisition, L.M.-P. and R.V. All authors have read and agreed to the published version of the manuscript.

**Funding:** This research is associated with Deanship of Scientific Research at King Khalid University, Kingdom of Saudi Arabia for funding this work through General Research Project under the grant number (RGP. 1/262/42).

**Data Availability Statement:** Experimental data is available upon request.

**Conflicts of Interest:** The authors declare no conflict of interest.

## Abbreviations

| | |
|---|---|
| $\theta_g$ | grid angle |
| $\Delta_{Lie}$ | error of inductor current |
| $f_s$ | Switching Frequency |
| $i_{act}$ | Actual Current |
| $i_L$ | Inductor Current |
| $i_{Lref}$ | Reference current |
| $i_{Lact}$ | Actual Current |
| $i_{ref}$ | Reference Current |
| $V_{CM-DM}$ | Total Common-Mode Voltage and Differential Mode Voltage |
| $V_{PV}$ | Output Voltage Of PV |
| $V_{TCM}$ | Transformer less Common-Mode Voltage |
| $CMV(V_{CM})$ | Common Mode Voltage |
| CPV | Parasitic Capacitance |
| DB-HBI | Dual-Buck Half-Bridge Inverter |
| $DMV(V_{DM})$ | Differential Mode Voltage |
| FWD | Free Wheeling Diode |
| HCC | Hysteresis Current Controls |
| IGBT | Insulated Gate Bipolar Transistor |
| MLI | Multi-Level Inverter |
| MPPT | Maximum Power Point Tracking |
| MOSFET | Metal Oxide Semiconductor Field Effect Transistor |
| NEC | National Electrical Code |
| NPC | Neutral Point Clamped |
| PLL | Phase Locked Loop |
| P&O | Perturb & Observe |
| SDPF | Switching Device Power Factor |
| TL | Transformer Less |
| TNP | T' Type NPC |
| ZCD | Zero-Crossing Detection |

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
