# Peer review of "Neutral Point Clamped Transformer-Less Multilevel Converter for Grid-Connected Photovoltaic System"

_electronics, doi:10.3390/electronics10080977_

Round 1

Reviewer 1 Report

The authors proposed a new topology (TL-TNP-MLI) for grid-connected PV system and tried tp demonstrate its potential through simulation and experiment. The following issues need to be adequately addressed before consideration for publication:

  1. The research objective is not clear. The authors need to further highlight the significance of the research.
  2. Little quantitative results and comparative analysis were provided to benchmark the improvement of the proposed topology compared with conventional ones. 
  3. The manuscript needs extensive language editing. 
  4. Figures need to have legend to indicate the physical representation of the symbols in figures. Also, straight line with arrow is recommended to provide additional narrative in figures instead of using dashed circular arrows. 

Author Response

Authors Reply:

       Dear reviewer, thank you so much for your appreciation. As per your suggestions, the presentation of the paper is improved.

      Your expert comments and suggestions to improve the quality of the manuscript are valuable for authors like us. We have made all the requested changes and added the new information, as given below in our response to your individual comments.

  • Note: The corrections in the revised manuscript are depicted in Blue color

The attachment has DETAILED reviewers response 

Reviewer 2 Report

An interesting publication concerns the cooperation of PV with the electric grid. In many countries, the power grids and infrastructure are often not adapted to the operation of larger amounts of PV, and as a result, PV panels are not fully used. Below are the comments for the publication:
Page 3. References should be cited in ascending order. [23, 25] [24] ??
Page 4. Arrange the pictures so that the arrows are clearly marked.
Page 6. [31-28] ?? Check citation.
There is no literature [32]. Similarly [36].
A nomenclature with a description of the symbols used together with units should be added to the publication.

It is worth considering the calculation of errors for the analysis.

Author Response

Authors Reply:

       Dear reviewer, thank you so much for your appreciation. As per your suggestions, the presentation of the paper is improved.

      Your expert comments and suggestions to improve the quality of the manuscript are valuable for authors like us. We have made all the requested changes and added the new information, as given below in our response to your individual comments.

  • Note: The corrections in the revised manuscript are depicted in blue color

The attachment has DETAILED reviewers response 

Reviewer 3 Report

The article is well written and interesting. The study is well reported, with an abundance of details and illustrations. However, the authors should consider addressing the following concerns to make the manuscript more aligned with a journal paper standard, but not a lab report. 

In the introduction/motivation section, why doing three-level, but not higher levels? A justification is expected. 

Table 4 presents the very important results comparing with other people's work. Are there any limitations or disadvantages of the proposed inverter in this work? It is expected that more interpretation of the table is added in the discussion, which can stimulate future research opportunities. 

There are a number of states of the arts recommended for enhancing the literature survey and the comparison of the results, for instance, 

F. Sebaaly, H. Y. Kanaan and N. Moubayed, "Three-level neutral-point-clamped inverters in transformerless PV systems — State of the art," MELECON 2014 - 2014 17th IEEE Mediterranean Electrotechnical Conference, Beirut, Lebanon, 2014, pp. 1-7, doi: 10.1109/MELCON.2014.6820496.

C. Anandababu and B. G. Fernandes, "A novel neutral point clamped transformerless inverter for grid-connected photovoltaic system," IECON 2013 - 39th Annual Conference of the IEEE Industrial Electronics Society, Vienna, Austria, 2013, pp. 6962-6967, doi: 10.1109/IECON.2013.6700287.

J. Li, "Design and Control Optimisation of a Novel Bypass-embedded Multilevel Multicell Inverter for Hybrid Electric Vehicle Drives," 2020 IEEE 11th International Symposium on Power Electronics for Distributed Generation Systems (PEDG), Dubrovnik, Croatia, 2020, pp. 382-385, doi: 10.1109/PEDG48541.2020.9244313.

A. Kadam and A. Shukla, "A 5-level high efficiency low cost Hybrid Neutral Point Clamped transformerless inverter for grid connected photovoltaic application," 2018 IEEE Applied Power Electronics Conference and Exposition (APEC), San Antonio, TX, 2018, pp. 3189-3194, doi: 10.1109/APEC.2018.8341558.

Other minor comments:
Keywords terms can be shortened - there is no need to include acronyms.  
The results illustrations are generally good. Figure 11 is too busy. Numbers and text are not readable without zooming in. 

Author Response

(The authors gave the same response as above.)

Round 2

Reviewer 3 Report

The response and revisions are satisfactory.

There are many English grammatical errors in the newly added paragraphs. 

For instance,

"Three-level NPC-MLI are versatilely used in motor control and PV grid connected applications compare to higher level MLIs."

"It is the main drawback of the NPC-MLI is maintain the DC-link capacitor balancing."

"There are variety of current controller discussed in the literature."

Please consider if using color for tables' column and row names is suitable for the journal. 

Author Response

Reviewer General Comments:

The response and revisions are satisfactory.

#Answer to the reviewer: Thanks for your appreciation. The author also thanks to reviewer for their valuable suggestion.

  1. . There are many English grammatical errors in the newly added paragraphs.

For instance,

"Three-level NPC-MLI are versatilely used in motor control and PV grid connected applications compare to higher level MLIs."

"It is the main drawback of the NPC-MLI is maintain the DC-link capacitor balancing."

"There are variety of current controller discussed in the literature."

#Answer to reviewer: 

Dear Reviewer, thank you for your observation. Authors apologize for the same.

In the revised paper the error has been corrected.

  1. Please consider if using color for tables' column and row names is suitable for the journal.

#Answer to reviewer: 

Dear Reviewer, thank you to the reviewer for their interest and suggestion.

In the revised paper, the format and the template are verified and corresponding correction are made.

Authors’ once gain thanking the reviews suggestions to improve the quality of the manuscript.
